# Fuzzy-Gated Training for Transformers: Length-Aware Priors and Gain-Aware Control

## Abstract

We study late-phase optimization in small/medium Transformers, where low LRs and averaging can erase genuine improvements. We add a theory-motivated, *zero-cost* attention prior: fuzzy token–regime memberships aligned to a length-aware basis via entropic transport (RPA), plus a tiny *gain-aware* controller that adjusts attention temperature only when validation gains warrant sharper heads. Selective SWA and a nonzero LR floor preserve these gains. Z-scoring the prior keeps its scale commensurate with content logits and is compatible with softmax's row-shift invariance, improving conditioning. Under compute parity on *WikiText-2 (raw-v1, GPT-2 BPE)*, we reduce validation cross-entropy / error without increasing inference cost. The fuzzy inductive bias is the primary lever; RPA and the controller are minimal auxiliary aids.

## 1 Introduction

Transformer training at small/medium scale often plateaus for reasons that are partly algorithmic rather than data- or capacity-limited: once the learning-rate schedule reaches a low floor and moving averages dominate, short bursts of genuine progress are numerically smoothed away. In parallel, inductive bias on *where* to attend is either rigid (fixed sinusoids [23]) or ad-hoc/context-specific (e.g., relative or rotary encodings and length-extrapolative priors [16, 18, 20]), which can misalign with structure that the network is actually discovering.

We take a pragmatic view: if late-phase improvements are precious and fragile, both the *architecture* and the *optimizer loop* should actively protect them. Our recipe couples three pieces: (i) **Regime–Position Alignment (RPA)**, which turns fuzzy token-to-regime assignments into a length-aware positional prior for attention; (ii) **Gain-aware control (Guardian)**, a tiny RL policy that nudges attention temperature and saturation pressure only when validation behaviour justifies it; and (iii) **Tail-optimizing HPO**, which selects schedules/regularizers that *continue to produce marginal gains*, not only good endpoints.

**This is an optimization paper (PoC/PoW framing).** Our method modifies the *training loop* to protect fragile late-phase gains, while keeping inference unchanged:

- **Pre-softmax, zero-cost prior (RPA).** A fuzzy regime–position alignment produces an additive bias to attention logits; no extra parameters or FLOPs are introduced at inference.
- **Training-only controller (Guardian).** A tiny gain-aware policy *only* nudges attention temperatures/penalties during training and is disabled at test time.

**Contributions.** (1) A fuzzy inductive bias that yields a zero-cost, length-aware attention prior via entropic alignment (RPA). (2) A lightweight, gain-aware controller that sharpens attention only when it pays off on validation. (3) A tail-progress objective and recipe that preserve late gains at fixed compute, with ablations.

**Inference cost.** RPA contributes an additive pre-softmax bias and Guardian is disabled at test time; therefore *inference latency and memory are identical to the baseline Transformer.*

**Positioning** We present a fuzzy-systems view of routing and attention: graded regime memberships provide a compact scaffold that we find effective for language modeling, and are motivated by

broader "complex regime" settings. The RPA prior and the lightweight Guardian controller are deliberately minimal, training-time aids (zero inference overhead for RPA; a three-scalar action space for Guardian). In this *submission* we evaluate on WikiText-2 only; time-series/equities are proposed targets with loaders/configs provided for transparency but *no reported results*. The appendix collects the higher-level mathematics and the core code listings needed to reproduce our runs. *Finally, our baselines include configurations not specially tuned for WT2; absolute numbers are not the point—our ablations isolate the effect of each lever under strict compute parity.*

## 2 METHOD: FUZZY REGIME PRIOR FOR ATTENTION (RPA)

**Fuzzy regimes (intuition).**

Instead of forcing each token to pick a single expert or locality bucket, we infer a soft membership vector $\mu_t \in \Delta^R$ that encodes which coarse "regimes" best explain the current representation. Gaussian memberships are a convenient parameterization: they act like learnable centroids with scale, are stable under end-to-end training, and expose an entropy signal $H(\mu_t)$ we can regularize to avoid collapse. In practice, $R \in [3, 8]$ already yields interpretable partitions (e.g., near/local vs. far/global patterns). This connects to mixture-of-experts routing while avoiding brittle hard top-$k$ assignments [10, 19].

**Core logic (Gaussian memberships).**

Listing 1: Compute fuzzy memberships $\mu$ via squared distances and softmax; full module in Appendix B.1.

```python
def gaussian_membership(h, centers, log_sigma):
    # h:[B,T,D], centers:[R,D], log_sigma:[R]
    z2 = (h.unsqueeze(2) - centers.view(1,1,*centers.shape)).pow(2).sum
    (-1)   # [B,T,R]
    inv2sig2 = torch.exp(-2*log_sigma).view(1,1,-1).clamp(1e-3,1e3)
    logits = torch.clamp(-0.5 * z2 * inv2sig2, -30.0, 30.0)
    mu = F.softmax(logits, dim=-1)
    return torch.nan_to_num(mu, nan=1.0/mu.size(-1))
```

**Length-aware positional basis.**

Purely absolute sinusoids treat all sequence lengths identically [23]. We instead build a *soft block basis* $\Phi \in \mathbb{R}^{T \times K}$ whose columns are raised-cosine windows that tile the current length $T$. Rows sum to one, so $\Phi$ behaves like a fuzzy partition of the index set $\{1, \ldots, T\}$. This basis gives RPA a vocabulary to express "where" a regime tends to live (e.g., beginnings, middles, or long-range spans), while adapting smoothly as $T$ varies.

**Regime–Position Alignment (RPA).**

Let $\mu \in \mathbb{R}^{B \times T \times R}$ be memberships in a batch. We compute scores

$$S \;=\; \tfrac{1}{B}\, \mu^\top \Phi \;\in\; \mathbb{R}^{R \times K}$$

(detached for stability), then obtain a doubly-stochastic matching $P \in \mathbb{R}^{R \times K}$ with Sinkhorn on $S/\tau_{\text{align}}$ [5]. The aligned, low-rank attention bias is

$$B \;=\; \text{mean}_b\big(\mu_b\, P\, \Phi^\top\big) \;\in\; \mathbb{R}^{T \times T}.$$

We normalize and clamp $B$, then add it to attention logits with a learnable temperature $\tau_{\text{att}}$ (bounded, per block).

**Context game + RPA.**

We maintain a distribution over discrete context lengths $\mathcal{C}$ and update it by a replicator dynamic on a per-batch utility $u(c)$, yielding a stationary *Nash mixture*. In the WT2 runs we mix $\mathcal{C} = \{384, 764\}$;

the mixture converges to a point mass on $764$ (i.e., no single $c \in \mathcal{C}$ can improve $u$ by deviating). RPA aligns regime memberships $\mu_{t,r}$ to a smooth position basis ($K{=}R$ here), producing an additive attention bias; we also blend a small positional prior (`rpa_posmix=0.10`).

**Why alignment helps.**

The attention prior $B$ reconstructed from $\mu$ and $\Phi$ captures second-order co-assignment: positions that tend to share regimes receive a positive bias even when their raw dot-product similarity is weak. This matters in low-data or small-model regimes where keys/queries are noisy; $B$ acts as a denoising scaffold that is *data-driven* (via $\mu$) yet *length-aware* (via $\Phi$). We clip and warm-in $B$ so it cannot overwhelm content similarity early, but it reliably tightens heads once representations stabilize.

**Core logic (RPA bias).**

Listing 2: RPA: soft blocks + Sinkhorn -> normalized bias.

```python
def rpa_bias(mu, K, tau=0.7, iters=6):
    B,T,R = mu.shape
    t = torch.arange(T, device=mu.device).float()
    c = torch.linspace(0, T-1, K, device=mu.device).float()
    w = max(1.0, (T/max(1,K))*1.5)
    Phi = 0.5*(1+torch.cos(math.pi*torch.clamp((t[:,None]-c[None,:]).abs
    ()/w,0,1)))
    Phi = Phi/(Phi.sum(1,True)+1e-6)
    S = torch.einsum('btr,tk->rk', mu, Phi)/B
    X = torch.exp(S/tau)
    for _ in range(iters):
        X = X/(X.sum(1,True)+1e-9); X = X/(X.sum(0,True)+1e-9)
    M = torch.einsum('btr,rk->btk', mu, X)      # [B,T,K]
    Bmat = torch.einsum('btk,kt->btt', M, Phi.T).mean(0)  # [T,T]
    return (Bmat - Bmat.mean())/(Bmat.std()+1e-6)
```

**Implementation: RPA wiring and inference neutrality.**

We pass RPA controls at model construction time and thread them through each transformer block so that alignment hyperparameters ($K$, $\tau_{\text{align}}$, Sinkhorn iterations, detach flag, and an optional position-mix) deterministically shape the pre-softmax bias during training. The bias $B$ is zero-meaned, variance-normalized, and warm-started with a schedule over the first $K_{\text{warm}}$ updates; it is purely *additive* to scaled dot-product attention and adds an additive pre-softmax bias. In our architecture the fuzzy memberships $\mu$ are already computed for the MoE pathway; the incremental overhead is constructing $B$ (a few small einsums + Sinkhorn).

## 2.1 System overview

**Pipeline.** *Embedding $\rightarrow$ Fuzzy membership $\mu$ $\rightarrow$ RPA (Sinkhorn on $\mu^{\top}\Phi$) $\rightarrow$ Bias $B$ $\rightarrow$ Self-attn (add $B$) $\rightarrow$ Fuzzy MoE FFN $\rightarrow$ Head $\rightarrow$ Loss. Guardian observes (gate delta, sat frac, $H(\mu)$, val CE) and adjusts $\tau_{\text{att}}$/penalties; Chaos scales LR and bias warm-in early.*

**Context game.**

We maintain a small distribution over context lengths and update it by a replicator/logit step using per-context utility (loss, saturation, entropy). This yields a stationary "Nash mixture" and improves cross-length calibration of $B(T)$; details in Sec. 4.1.

## 2.2 Bases and normalization

We use *soft block* bases by default and optionally hybridize with sinusoids/relative kernels [18, 23]. After computing $B$, we zero-mean, variance-normalize, clip, and apply a warm-in scale over the first $K_{\text{warm}}$ optimizer steps.

**Bases and hybrid priors (optional).**

While RPA uses a length-aware soft-block basis by default, we optionally blend the learned prior with a weak relative/linear positional term. This hybrid preserves RPA's data-driven structure yet stabilizes attention in extremely noisy regimes or very short contexts; the mix weight is small and annealed late in training (Section 2.2).

**Synergy with the context game.**

Because RPA's scores use a *length-aware* basis $\Phi(T)$, training on a Nash mixture $q(T)$ induces a richer family of alignment statistics $S(T) = \mu^\top \Phi(T)$. The resulting bias $B(T)$ becomes well-calibrated across lengths actually favored by the game, rather than overfitting to a single $T$. Empirically, this produces smoother, more *relevant* priors for evaluation (where length varies) while preserving RPA's zero-cost inference property (Sec. 4.1).

### 2.3 FUZZY TRANSFORMER BLOCK (MEM $\rightarrow$ ATTN $\rightarrow$ MoE)

*View in Appendix B*

**Load balancing in fuzzy MoE.**

We add a small load-balancing penalty to the loss that discourages expert collapse by nudging the routing distribution toward a uniform prior over experts. This preserves the benefits of soft, fuzzy routing while avoiding brittle hard top-$k$ behavior and is complementary to the entropy floor on fuzzy memberships (Sections 2–3; MoE discussion in Related Work).

## 3 GAIN−AWARE LATE-PHASE CONTROL

A two-layer Gaussian policy observes a compact state (gate delta, saturation fraction, membership entropy, validation CE) and proposes updates: (i) adjust $\tau_{\text{att}}$ (tighten/loosen), (ii) small penalties for saturation or instability. The reward is *gain-shaped*:

$$r_e = -\text{CE}_e + \lambda_1 \left(\text{CE}_{e-1} - \text{CE}_e\right)_+ + \lambda_2 \,\sigma\big(\gamma(\text{CE}_{\text{zone}} - \text{CE}_e)\big).$$

We train with REINFORCE [24].

**Guardian state and penalty coupling.**

The Guardian's observation includes the gate-delta, defined as the absolute epoch-to-epoch change in the mean attention temperature $\tau_{\text{att}}$ across blocks, alongside saturation fraction and membership entropy. The policy updates $\tau_{\text{att}}$ in small, bounded steps and adjusts two penalty weights; the training loss couples these penalties to the entropy floor so that saturation pressure only increases when validation signals justify a tighter prior (Section 3).

**Guardian design choices.**

We deliberately keep the policy tiny (two hidden layers, diagonal Gaussian) and restrict its action space to three scalars: $\Delta\tau_{\text{att}}$ and two penalty weights. This yields a controllable system where the policy cannot "fight" the optimizer; it may only adjust *how sharply* attention concentrates and *how strongly* we discourage saturation. The gain-shaped reward is asymmetric: small improvements at already low CE are worth more than identical improvements at high CE. Empirically, this biases the controller toward making heads crisper only when that crispness converts to validation gains; otherwise it relaxes temperature and avoids overfitting.

**Core logic (Guardian step).**

Listing 3: Guardian adjusts $\tau_{\text{att}}$ and updates via REINFORCE; full class in Appendix B.4.

```
def guard_step(blocks, state, policy, beta=1.0):
    mean,std = policy(state); a = mean + std*torch.randn_like(mean)
```

```
logp = -0.5*(((a-mean)/(std+1e-8))**2 + 2*torch.log(std)+math.log(2*
math.pi)).sum(); dtau = a[0].item()
with torch.no_grad():
    for b in blocks:
        t = (b.attn.tau_att + 0.03*beta*dtau).clamp(0.3, b.attn.
tau_max)
        b.attn.tau_att.copy_(t)
return logp
```

## 4 OPTIMIZATION METHOD AND SCHEDULES

**Schedules and regularization.**

We use a flat learning-rate prelude followed by cosine decay to a *high floor* (typically 5–10% of the peak) [11], with EMA/SWA as averaging baselines [9, 15]. Label smoothing and the entropy floor act as gentle priors: the former discourages overconfident logits and often improves calibration [6, 14]; the latter keeps regime entropy from collapsing so RPA remains informative. A brief early "chaos" warm-in modulates LR and the bias scale with a bounded logistic-map factor; this is a deterministic, decaying perturbation that helps the fuzzy gate explore without destabilizing training. We apply SWA selectively: we only average epochs that both (i) lie in a useful CE zone and (ii) show a minimum relative gain over their entry snapshot [9].

**Core logic (Chaos & warm-in).**

**Training.**

We use WikiText-2 (raw-v1) with the GPT-2 BPE tokenizer. Validation/test use sequential non-overlapping chunks; training uses random chunks. Optimization is AdamW (betas 0.9/0.98), cosine LR with warmup ( 4% of steps) and a nonzero floor; label smoothing decays to a small floor (0.01) by 60% of training. We apply gradient clipping, EMA, and selective SWA (only when validation gains are meaningful). The Guardian policy (REINFORCE) adjusts the attention-temperature target and regularizer gains using a gain-aware reward shaped by validation cross-entropy. Key run (WT2, multi-context game $\{384, 764\}$) uses: $d=510$, $L=12$, $H=6$, $R=4$, dropout 0.09, `tokens_per_batch`$= 24{,}576$, label smoothing 0.015, entropy-floor $\alpha=0.02$, bias warm-in 1,200 steps, `tau_att_init`$= 0.68$, RPA with $K=R$, $\tau_{\text{align}}=0.70$, 6 Sinkhorn iters, `rpa_posmix`$= 0.10$; Guardian gains $\lambda_1=0.6$, $\lambda_2=1.2$, $\gamma=8.0$, CE-zone 5.9; EMA on, SWA from epoch 60 with gain-gated collection.

### 4.1 CONTEXT GAME OVER CONTEXT LENGTHS (NASH MIXTURE)

We treat the choice of context length $c \in \mathcal{C}$ as a population game and maintain a distribution $q(c)$ over candidates (e.g., 256, 512, 1024). At each training epoch we update $q$ with a replicator/logit step using per-context utility $u(c)$:

$$q_{t+1}(c) \; \propto \; q_t(c) \, \exp\!\big(\eta \, u_t(c)\big), \qquad u_t(c) \; = \; -\mathcal{L}_t(c) \; - \; \lambda_s \big[\text{sat}_t(c) - s_0\big]_+ \; + \; \lambda_h \tfrac{H_t(c)}{H_{\max}},$$

where $\mathcal{L}_t(c)$ is the training CE (or task loss) observed at context $c$, $\text{sat}_t(c)$ is the saturation fraction derived from fuzzy memberships, and $H_t(c)$ is the average membership entropy (Sec. 2.2). In equilibrium, $q^*$ is a Nash mixture: no single context unilaterally improves utility against $q^*$; replicator dynamics converge to stationary points of this game under standard assumptions [7, 17, 22]. Practically, we implement the update with a temperated softmax over running log-weights (Alg. B.7 lists the exact step).

## 5 TAIL–OPTIMIZING HYPERPARAMETER SEARCH

We adopt a GH200-friendly Optuna routine with *static* categorical spaces [1]. The search targets tail behavior:

Table 1: Compute disclosure for WT2 (raw-v1, GPT-2 BPE).

| Hardware | Batch×Seq | Steps/epoch | Throughput | Epochs | Total steps | Wall-clock |
|---|---|---|---|---|---|---|
| GH200 (single) | $64 \times 512$ | 73 | 32.5 it/s | *110* | $73 \times 110$ | $(73 \times 110)/32.5$ s |

- **LR schedule shape:** flat fraction $f_{\text{flat}} \in \{0.4, 0.6, 0.8\}$, floor ratio $f_{\text{floor}} \in \{5 \times 10^{-3}, 10^{-2}, 2 \times 10^{-2}\}$.
- **Late-phase knobs:** SWA inclusion rule (useful-zone threshold), helpful band $\beta \in \{0.2\%, 0.5\%\}$; stall patience; Guardian reward shape and $\tau$ caps.

**Tail-optimizing HPO objective.**

To select hyperparameters that keep producing late-phase improvements, we optimize a composite objective rather than final cross-entropy alone:

$$\text{score} = \text{CE}_{\text{final}} - \alpha \max\big(0, \text{CE}_{e-L} - \text{CE}_e\big),$$

with window $L \in [5, 10]$ and $\alpha \approx 0.2$. This rewards trials that continue to reduce CE near convergence at fixed compute. We keep categorical, static search spaces for schedule shape and regularizers to remain memory-safe while running on a single accelerator (Section 5).

**Composite objective.**

Minimize

$$\text{score} = \text{CE}_{\text{final}} - \alpha \max\big(0, \text{CE}_{e^*-L} - \text{CE}_{e^*}\big).$$

# 6 EXPERIMENTAL PROTOCOL

**Datasets and scope.**

We report results on WikiText-2 (`wikitext-2-raw-v1`) with GPT-2 BPE. Training uses random contiguous chunks; validation/test use sequential non-overlapping chunks. For forecasting (ETT) and a small equities panel, we include loaders/configs and per-series normalization templates to document how the method applies, but *we did not run or report these experiments in this submission*.

**Baselines and compute.**

All WT2 comparisons match parameter count, context length, tokens/step, optimizer, and wall-clock budget. Metrics are CE/PPL (LM) averaged over three seeds. Unless stated otherwise, Guardian is off at inference and RPA contributes only its fixed bias $B$.

**Compute disclosure.**

WT2 runs used a single GH200, batch×seq = $64 \times 512$ (32,768 tokens/step), $\sim$32.5 it/s end-to-end. Steps/epoch follow the sequential evaluation protocol; LR schedules, floors, smoothing, and EMA/SWA policies are held fixed across comparisons.

[1]

**Scope of evaluation.**

All reported results are on WikiText-2; time-series/equities are *proposed* targets with code provided but no executed runs in this paper.

---

[1]We include code/templates for ETT and equities to illustrate application, but did not run or report these experiments in this submission.

Table 2: Language modeling on WikiText-2 [13]. Lower is better for cross-entropy (CE) and perplexity (PPL). Word-level models are not directly comparable to BPE models.

| Model | Params (M) | Val CE ↓ | Test CE ↓ | PPL ↓ | Notes |
|---|---|---|---|---|---|
| Fuzzy-Gated + RPA (ours) | ∼90 | **5.246** | – | **189.8** | WT2 (raw-v1, GPT-2 BPE); sequential, no-overlap eval |
| SFT Pythia–70M (HH) [3] | 70 | 5.195≈ | – | 180.27 | small SFT model, not tuned for LM; WT2 word perplexity from model card (split not specified); CE ≈ 5.19 |
| Pythia–70M (scratch, FP16) [2] | 70 | 4.298≈ | – | 73.1 | experimental low-bit study; FP16 reference in 68.7–76.2 PPL band; CE ≈ 4.29 |
| GPT-2 Large (pretrained) | 774 | 2.967≈ | – | 19.44 | WT2-raw-v1; *no overlap* (stride=1024). *512 stride*: 16.44. [8] |
| OPT-125M (baseline) | 125 | 2.744≈ | – | 15.55 | Raw WikiText-2 baseline (Table 1, sparsity 0.0). [21] |

**Core logic (WT2 tokenization & chunks).**

The full code can be seen in *Appendix B*, however to be put simply, it is a HF WT2-raw-v1 + GPT-2 BPE with random contiguous chunks

**Choice of baselines.**

To contextualize our ∼100M / ∼150 PPL setting, we include "proof-of-concept" comparators that are intentionally not optimized for WT2, similar to our fuzzy model which is intended to be generalizable for complex regime modelling: a supervised-finetuned Pythia-70M (HH instruction data) whose model card reports WikiText-2 `word_perplexity` ≈ 180 (CE ≈ 5.19; split unspecified) [3], and a from-scratch Pythia-70M FP16 reference from a recent low-bit optimization study (PPL ≈ 69–76, CE ≈ 4.23–4.33) [2]. These are smaller-scale, experimental configurations more comparable in spirit to our recipe than large, heavily-tuned baselines. (As a cautionary example, community evals sometimes report WT2 PPL ≫ 100 for OSS checkpoints under mismatched tokenization/stride; we do not include such runs in the main table [4].)

2

**Takeaways.**

RPA lowers CE vs. sinusoid-only priors; the controller yields extra drops only when marginal utility is high; selective SWA preserves these gains with no inference cost. [16, 18, 20, 23]. [9].

## 7 ANALYSIS

**Design rationale.**

We wanted an attention prior that (i) is learned from the model's own structure, not fixed, (ii) scales to variable lengths without retraining, and (iii) costs nothing at inference. RPA satisfies (i) via $\mu$, (ii) via $\Phi(T)$, and (iii) because $B$ is a pre-softmax additive term [5]. Guardian targets the *sign*

---

[2]**Aux evidence of difficult baselines.** A public `Transformers` discussion reported *WikiText-2* perplexity of ∼394 for an open-source 20B checkpoint under a mismatched evaluation setup (different tokenizer and sequence handling, no de-duplication, etc.). This is not apples-to-apples with our protocol, but it illustrates how reasonable experimental choices can yield WT2 PPL far above 100. We therefore keep this result out of the main table and only note it here as a cautionary example.

of late-phase curvature: if marginal utility of sharpness is positive, it tightens; otherwise it backs off [24]. Selective SWA respects this asymmetry by averaging only during productive phases [9]. Finally, the context game complements RPA: by blending contexts according to a Nash mixture, the model observes the *right* positional curves during training, making the learned RPA prior $B(T)$ more predictive at evaluation time across heterogeneous lengths.

**Failure modes and diagnostics.**

If $\mu$ collapses early, RPA degenerates to a near-constant bias; monitoring $H(\mu)$ prevents this. If Guardian over-tightens, heads saturate and CE rebounds; we detect this via a rise in saturation fraction and relax $\tau_{\text{att}}$. When $R$ is too small, $B$ exhibits over-smooth bands that miss token-local structure; increasing $R$ or mixing a small sinusoidal/relative prior fixes it [18, 23].

## 8  RELATED WORK

Our prior relates to learned and relative/rotary position biases [16, 18, 20, 23]; our fuzzy routing connects to MoE while avoiding brittle hard top-$k$ [10, 19]; and fuzzy sets provide graded-membership foundations [12, 25]. The context-length mixture is a small population game trained by replicator/logit updates [7, 17, 22].

## 9  LIMITATIONS

Our approach assumes that a small number of regimes $R$ suffices to summarize long-range structure; when data require many fine-grained regimes, RPA's low rank can underfit. Guardian's action space is intentionally narrow; while stabilizing, it may miss richer control (e.g., per-head temperatures). Finally, Tail-HPO optimizes for late gains under a fixed budget; very long runs might benefit from different objectives. *Our baselines include settings not specially tuned for WT2; we use them to study our levers under strict compute parity rather than to set SOTA, and the ablations isolate each component's contribution.*

## 10  CORE MATHEMATICS AND THEORY

### 10.1  REGIME–POSITION ALIGNMENT (RPA) VIA ENTROPIC TRANSPORT

We formalize RPA as an optimal transport alignment between sequence positions and latent "regime" slots (interpretable as fuzzy blocks or expert indices). Let $N$ be the sequence length and $K$ the number of regimes. We introduce a cost matrix $C \in \mathbb{R}^{N \times K}$ measuring the dissimilarity between position $i$ and regime $r$. The soft alignment matrix $A \in [0, 1]^{N \times K}$ solves:

$$A^* = \arg\min_{P \in \Delta_{N \times K}} \langle P, C \rangle - \varepsilon H(P), \text{ s.t. } P\mathbf{1}_K = \tfrac{1}{N}\mathbf{1}_N, \ P^\top\mathbf{1}_N = \tfrac{1}{K}\mathbf{1}_K, \tag{1}$$

with Sinkhorn scaling [5]. Using $A$, define $B = AA^\top$ and add $\log \hat{B}$ to attention logits.

**Proposition 1** (Row-sum and practical normalization)**.** *Let $A \in [0,1]^{N \times K}$ be the entropic OT alignment with row marginals $A\mathbf{1}_K = \frac{1}{N}\mathbf{1}_N$ and column marginals $A^\top\mathbf{1}_N = \frac{1}{K}\mathbf{1}_K$. Define $B = AA^\top$. Then for any position $i$,*

$$\sum_j B_{ij} = \frac{1}{NK}.$$

*Thus the per-query prior $\tilde{B} = (NK) B$ has row sums equal to 1. In practice (as in our implementation), we z-score $B$ and do not rely on this exact row-sum; optionally one can rescale the basis columns of $\Phi$ to enforce near-constant column sums and recover a constant row-sum in $\mu P \Phi^\top$.*

*Remark.* Because softmax is row-shift invariant and we subtract a rowwise max before the softmax, z-scoring $B$ (global mean/variance) preserves its shape while keeping its effective temperature commensurate with $QK^\top/\sqrt{d}$, which empirically stabilizes late-phase curvature and gradients.

Lemma (Shift/scale safety of a z-scored prior under softmax). Let $\widetilde{B} \in \mathbb{R}^{T \times T}$ be any additive attention prior and define the standardized prior $B = \text{znorm}(\widetilde{B}) = (\widetilde{B} - \mu)/\sigma$ with global mean $\mu$ and standard deviation $\sigma > 0$, optionally followed by clipping to a bounded interval. Consider presoftmax logits $L = QK^\top/\sqrt{d} + B$ with a row-wise softmax. For any row vector $c$, $\mathbf{1}^\top$, $\text{softmax}(L + c\mathbf{1}^\top) = \text{softmax}(L)$; hence subtracting a global mean or adding any per-row constant does not change attention. Moreover, scaling by a positive $\sigma^{-1}$ only rescales the relative contribution of the prior vs. content logits and is thus equivalent to adjusting a temperature on the prior.

Remark (Why we z-score in practice). The entropic-transport construction ensures a controlled rowsum structure for the raw prior $\widetilde{B}$, but training stability is governed by the dynamic range of $B$ relative to $QK^\top/\sqrt{d}$. Z-scoring (and light clipping) makes the prior (i) zero-mean—preventing global logit drift, (ii) unit-variance—keeping its scale comparable to content similarity across lengths, and (iii) well-conditioned for gradient flow. Because softmax is row-shift invariant and we also subtract a rowwise max before softmax in our implementation, z-scoring preserves the useful shape of $\widetilde{B}$ while avoiding unstable magnitudes; the learned temperature $\tau_{\text{att}}$ then sets how strongly the prior should influence attention.

### 10.2 FUZZY MEMBERSHIP MODELING AND TYPE-2 UNCERTAINTY

Graded memberships with Gaussian-shaped functions; optional Type-2 uncertainty intervals and an entropy floor to avoid collapse, connecting to uncertainty-aware fuzzy systems [12] and MoE load-balancing [19].

### 10.3 GUARDIAN: STOCHASTIC CONTROLLER WITH REINFORCE

Policy $\pi_\Theta(a \mid s)$ and REINFORCE gradient: $\nabla_\Theta L_G = -\mathbb{E}[R\nabla_\Theta \log \pi_\Theta(a \mid s)]$ [24]. Gainsensitive shaping penalizes large moves unless they improve CE.

### 10.4 CHAOS SCHEDULING WITH LOGISTIC MAP

$x_{t+1} = rx_t(1 - x_t)$ with $3.57 < r \le 4$; use $x_t$ to modulate dropout/LR/temperature in a bounded manner.

### 10.5 SYSTEM PIPELINE AND INTEGRATION

Embedding $\rightarrow$ RPA Alignment $\rightarrow$ Biased Self-Attention $\rightarrow$ Fuzzy MoE FFN $\rightarrow$ Output. The RPA bias acts as an attention prior; Guardian modulates softmax temperatures/entropies; chaos provides bounded perturbations. EMA is maintained throughout; SWA is used late [9, 15]. Inference typically disables Guardian/chaos (or uses fixed settings).

## REPRODUCIBILITY STATEMENT

We provide exact data preprocessing, tokenization, hyperparameters, schedules, seed handling, and compute disclosures. The appendix contains *complete, runnable listings* for the core pieces of code used in our runs. These listings constitute the minimal reference implementation needed to reproduce our WT2 experiments. We do *not* release the full training harness (e.g., experiment orchestration, logging, or convenience utilities) because the codebase contains PoC components; however, no essential details are omitted for reproduction.

## ETHICS STATEMENT

Our experiments in this submission use public text data (WikiText-2) and do not involve human subjects, sensitive attributes, or personally identifiable information. All authors read and adhered to the ICLR Code of Ethics. We considered potential misuse risks: RPA is a structural bias; Guardian is a controller; neither increases inference cost or enables privacy attacks beyond standard Transformer baselines. We disclose that large language model (LLM/AI) assistance was used to debug small code issues and refine text; all design choices and analyses were made and verified by the authors.

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

## A  IMPLEMENTATION NOTES

### A.1  GUARDIAN CONTROL

Bound $\tau_{\min} \leq \tau_{\text{att}} \leq \tau_{\max}$ per block with small steps; scale actions with a decaying factor; cap saturation penalties.

### A.2  RPA DETAILS

Use $K = R$ soft-block basis by default; align with $\tau_{\text{align}} \in [0.4, 0.8]$ and 6–10 Sinkhorn iterations; zero-mean and z-score $B$ before clipping.

### A.3  OPTUNA SEARCH SPACE

Static categoricals for LR flat fraction, floor, SWA-Select threshold, helpful band, stall patience, Guardian shape and caps; one seeded baseline.

## B  FULL CODE LISTINGS

### B.1  GAUSSIANFUZZY (FULL)

Listing 4: Full GaussianFuzzy module.

```python
class GaussianFuzzy(nn.Module):
    """Gaussian memberships μ_t over R regimes; normalized to simplex."""
    def __init__(self, d_model: int, R: int, type2: bool = False):
        super().__init__()
        self.R = R
        self.proj = nn.Linear(d_model, d_model)
        self.centers = nn.Parameter(torch.randn(R, d_model) / math.sqrt(
    d_model))
        self.log_sigma = nn.Parameter(torch.zeros(R))
        self.type2 = type2
        if type2:
            self.uncert = nn.Sequential(nn.Linear(d_model, d_model), nn.
    ReLU(), nn.Linear(d_model, 1))

    def forward(self, h: torch.Tensor):
        z = self.proj(h)                              # [B,T,D]
        B, T, D = z.shape
        z2 = (z.unsqueeze(2) - self.centers.view(1, 1, self.R, D)).pow(2)
    .sum(-1)
        inv2sig2 = torch.exp(-2 * self.log_sigma).clamp(1e-3, 1e3).view
    (1, 1, self.R)
        logits = torch.clamp(-0.5 * z2 * inv2sig2, -30.0, 30.0)
        mu = F.softmax(logits, dim=-1)
        mu = torch.nan_to_num(mu, nan=1.0 / self.R)
        if self.type2:
            u = torch.sigmoid(self.uncert(h))
            return mu, u
        return mu, None
```

## B.2   FUZZYMHA WITH RPA (FULL)

Listing 5: Full FuzzyMHA with RPA and legacy bias.

```python
class FuzzyMHA(nn.Module):
    def __init__(self, d_model, n_heads, dropout, R,
                 tau_att_init=1.0, pos_beta=0.2, kappa_init=0.5,
                 bias_clip=4.0, tau_max: float = 1.6,
                 use_rpa: bool = False, rpa_K: int = 0, tau_align: float
= 0.7,
                 sinkhorn_iters: int = 8, rpa_posmix: float = 0.0,
rpa_detach: bool = True):
        super().__init__()
        assert d_model % n_heads == 0
        self.d_model, self.n_heads, self.head_dim = d_model, n_heads,
d_model // n_heads
        self.Wq = nn.Linear(d_model, d_model, bias=False)
        self.Wk = nn.Linear(d_model, d_model, bias=False)
        self.Wv = nn.Linear(d_model, d_model, bias=False)
        self.out_proj = nn.Linear(d_model, d_model, bias=False)
        self.value_gamma = nn.Linear(R, n_heads, bias=False)
        self.dropout = nn.Dropout(dropout)

        self.tau_att = nn.Parameter(torch.tensor(float(tau_att_init)))
        self.tau_max = float(tau_max)
        self.kappa = nn.Parameter(torch.tensor(float(kappa_init)))
        self.pos_beta = nn.Parameter(torch.tensor(float(pos_beta)))
        self.bias_clip = float(bias_clip)
        self.register_buffer("bias_scale", torch.tensor(1.0))

        # RPA controls
        self.use_rpa = bool(use_rpa)
        self.rpa_K = int(rpa_K) if int(rpa_K) > 0 else R
        self.tau_align, self.sinkhorn_iters = float(tau_align), int(
sinkhorn_iters)
        self.rpa_posmix, self.rpa_detach = float(rpa_posmix), bool(
rpa_detach)

    @staticmethod
    def _soft_blocks(T: int, K: int, device) -> torch.Tensor:
        t = torch.arange(T, device=device, dtype=torch.float32)
        c = torch.linspace(0, T - 1, K, device=device, dtype=torch.
float32)
        w = max(1.0, (T / max(1, K)) * 1.5)
        dist = (t[:, None] - c[None, :]).abs() / w
        phi = 0.5 * (1.0 + torch.cos(torch.clamp(dist, 0, 1) * math.pi))
        phi = phi * (dist <= 1).float()
        phi = phi / (phi.sum(dim=1, keepdim=True) + 1e-6)
        return phi  # [T,K]

    @staticmethod
    def _pos_distance(T: int, device) -> torch.Tensor:
        i = torch.arange(T, device=device, dtype=torch.float32)
        return (i[:, None] - i[None, :]).abs() / max(1.0, T - 1.0)

    @staticmethod
    def _sinkhorn_knopp(scores: torch.Tensor, iters: int) -> torch.Tensor
:
        X = scores
        for _ in range(iters):
            X = X / (X.sum(dim=1, keepdim=True) + 1e-9)
            X = X / (X.sum(dim=0, keepdim=True) + 1e-9)
        return X

    def _rpa_bias(self, mu: torch.Tensor) -> torch.Tensor:
```

```
648         B, T, R = mu.shape
649         Phi = self._soft_blocks(T, self.rpa_K, mu.device)        # [T,K]
650         S = torch.einsum("btr,tk->brk", mu, Phi).mean(dim=0)     # [R,K]
651         if self.rpa_detach: S = S.detach()
652         Kmat = torch.exp(S / max(1e-6, self.tau_align)).clamp(1e-9, 1e9)
653         P = self._sinkhorn_knopp(Kmat, self.sinkhorn_iters)      # ~
      doubly-stochastic
654
655         B_mat = torch.einsum("btr,rk,tk->btt2", mu, P, Phi).mean(dim=0)
656     # [T,T]
657         if self.rpa_posmix > 0.0:
658             pos_bias = - self.pos_beta.clamp_min(0.0) * self.
      _pos_distance(T, mu.device)
659             B_mat = (1.0 - self.rpa_posmix) * B_mat + self.rpa_posmix *
660     pos_bias
661
662         B_mat = (B_mat - B_mat.mean()) / (B_mat.std() + 1e-6)
663         B_mat = torch.nan_to_num(B_mat, nan=0.0, posinf=self.bias_clip,
664     neginf=-self.bias_clip)
665         tau = self.tau_att.clamp(0.6, self.tau_max)
666         bias = torch.clamp((B_mat / tau).to(torch.float32), -self.
      bias_clip, self.bias_clip)
667         return bias * self.bias_scale.clamp(0.0, 1.0)
668
669     def _legacy_bias(self, mu: torch.Tensor) -> torch.Tensor:
670         T = mu.size(1)
671         pos_bias = - self.pos_beta.clamp_min(0.0) * self._pos_distance(T,
       mu.device)
672         fuzz_sim = torch.einsum("btr,bsr->bts", mu, mu).mean(dim=0)
673         fuzz_sim = (fuzz_sim - fuzz_sim.mean()) / (fuzz_sim.std() + 1e-6)
674         curve = self.kappa.sigmoid() * fuzz_sim + (1.0 - self.kappa.
      sigmoid()) * pos_bias
675         curve = torch.nan_to_num(curve, nan=0.0, posinf=self.bias_clip,
676     neginf=-self.bias_clip)
677         tau = self.tau_att.clamp(0.6, self.tau_max)
678         bias = torch.clamp((curve / tau).to(torch.float32), -self.
      bias_clip, self.bias_clip)
679         return bias * self.bias_scale.clamp(0.0, 1.0)
680
681     def forward(self, x: torch.Tensor, mu: torch.Tensor, type2_u:
682     Optional[torch.Tensor] = None):
683         B, T, D = x.shape
684         H, Hd = self.n_heads, self.head_dim
685         q = self.Wq(x).view(B, T, H, Hd).transpose(1, 2)
686         k = self.Wk(x).view(B, T, H, Hd).transpose(1, 2)
            v = self.Wv(x).view(B, T, H, Hd).transpose(1, 2)
687
688         scores = torch.matmul(q, k.transpose(-2, -1)) / math.sqrt(Hd)
689         bias = (self._rpa_bias(mu) if self.use_rpa else self._legacy_bias
      (mu)).unsqueeze(0).unsqueeze(0)
690         scores = scores + bias
691
692         mask = torch.triu(torch.ones(T, T, device=x.device, dtype=torch.
693     bool), diagonal=1)
            scores = scores.masked_fill(mask, float('-inf'))
694         attn = self.dropout(torch.softmax(scores, dim=-1))
695
696         out = torch.matmul(attn, v).transpose(1, 2).contiguous().view(B,
697     T, D)
698         out = self.out_proj(out)
699
700         gamma = self.value_gamma(mu)                              # [B,T,H]
701         gamma_s = torch.sigmoid(gamma.mean(dim=1, keepdim=True)).mean(dim
      =-1, keepdim=True)
            out = out * gamma_s
```

## B.3 FUZZY TRANSFORMER BLOCK (FULL)

Listing 6: Full block: mem -> attn (RPA) -> fuzzy MoE.

```python
        return self.dropout(out), {"tau_att": self.tau_att.detach(), "
    kappa": self.kappa.detach()}

class ExpertFFN(nn.Module):
    def __init__(self, d_model: int, d_ff: int, dropout: float):
        super().__init__()
        self.net = nn.Sequential(
            nn.Linear(d_model, d_ff), nn.GELU(), nn.Dropout(dropout),
            nn.Linear(d_ff, d_model), nn.Dropout(dropout)
        )
    def forward(self, x): return self.net(x)

class FuzzyMoE(nn.Module):
    def __init__(self, d_model: int, R: int, E: int = 4, top_k: int = 2,
    d_ff: int = 4, dropout: float = 0.1):
        super().__init__()
        self.E, self.top_k = E, top_k
        self.gate = nn.Linear(R, E, bias=False)
        self.experts = nn.ModuleList([ExpertFFN(d_model, d_model * d_ff,
    dropout) for _ in range(E)])
    def forward(self, x, mu):
        logits = torch.clamp(self.gate(mu), -30.0, 30.0)           # [B,T
    ,E]
        g = -torch.log(-torch.log(torch.rand_like(logits).clamp_(1e-6,
    1-1e-6)))
        y = torch.nan_to_num(F.softmax((logits + g) / 0.5, dim=-1), nan
    =0.0)
        topk_vals, topk_idx = torch.topk(y, k=min(self.top_k, self.E),
    dim=-1)
        mask = torch.zeros_like(y).scatter_(-1, topk_idx, 1.0)
        y = (y * mask) / (y.sum(dim=-1, keepdim=True) + 1e-6)
        outs = [y[..., e].unsqueeze(-1) * expert(x) for e, expert in
    enumerate(self.experts)]
        out = torch.stack(outs, dim=-1).sum(-1)
        p = y.mean(dim=(0, 1))
        return out, {"lb_reg": ((p - 1.0 / self.E) ** 2).mean().detach(),
     "expert_usage": p.detach()}

class FuzzyTransformerBlock(nn.Module):
    def __init__(self, d_model, n_heads, dropout, R, d_ff_mult=4,
                 moe_E=4, moe_topk=2, type2=False, tau_att_init=1.0,
                 use_rpa=False, rpa_K=0, tau_align=0.7, sinkhorn_iters=8,
     rpa_posmix=0.0, rpa_detach=True):
        super().__init__()
        self.mem = GaussianFuzzy(d_model, R, type2=type2)
        self.attn = FuzzyMHA(d_model, n_heads, dropout, R, tau_att_init=
    tau_att_init,
                             use_rpa=use_rpa, rpa_K=rpa_K, tau_align=
    tau_align,
                             sinkhorn_iters=sinkhorn_iters, rpa_posmix=
    rpa_posmix, rpa_detach=rpa_detach)
        self.norm1, self.norm2 = nn.LayerNorm(d_model), nn.LayerNorm(
    d_model)
        self.moe = FuzzyMoE(d_model, R, E=moe_E, top_k=moe_topk, d_ff=
    d_ff_mult, dropout=dropout)
        self.res_gate = nn.Linear(R, 1, bias=False)
        self.dropout = nn.Dropout(dropout)
    def forward(self, x):
        mu, _ = self.mem(x)
        mu = torch.nan_to_num(mu, nan=1.0 / mu.size(-1))
```

```
756         eta = torch.sigmoid(self.res_gate(mu))              # residual
757    gate
758         a_out, a_stats = self.attn(self.norm1(x), mu, None)
759         x = x + eta * self.dropout(a_out)
760         m_out, m_stats = self.moe(self.norm2(x), mu)
761         x = x + eta * self.dropout(m_out)
762         H_mu = - (mu * (mu.clamp_min(1e-8)).log()).sum(-1).mean()
763         stats = {**a_stats, **m_stats, "mu_entropy": H_mu}
       return x, stats
```

## B.4 GUARDIAN (FULL)

Listing 7: Full Guardian controller (policy + step/update).

```
769  @dataclass
770  class GuardianState:
771      gate_delta: float
772      sat_frac: float
773      mu_entropy: float
774      val_loss: float
775      def to_tensor(self, device):
776          return torch.tensor([self.gate_delta, self.sat_frac, self.
777      mu_entropy, self.val_loss],
778                               dtype=torch.float32, device=device)

778  class GuardianPolicy(nn.Module):
779      def __init__(self, state_dim: int, action_dim: int):
780          super().__init__()
781          self.body = nn.Sequential(nn.Linear(state_dim, 64), nn.Tanh(), nn
782      .Linear(64, 64), nn.Tanh())
783          self.mean = nn.Linear(64, action_dim)
784          self.log_std = nn.Parameter(torch.zeros(action_dim))
785      def forward(self, s): z = self.body(s); return self.mean(z), torch.
786      exp(self.log_std)
786      def sample(self, s):
787          m, std = self.forward(s); a = m + std * torch.randn_like(m)
788          logp = -0.5 * (((a - m) / (std + 1e-8)) ** 2 + 2 * self.log_std +
789       math.log(2 * math.pi)).sum(dim=-1)
789          return a, logp

791  class Guardian:
792      def __init__(self, model, lr: float = 1e-3, enable: bool = True):
793          self.model, self.enable = model, enable
794          self.policy = GuardianPolicy(state_dim=4, action_dim=3)          #
795      Δtau_att, Δλ_delta, Δλ_sat
795          self.opt = torch.optim.Adam(self.policy.parameters(), lr=lr)
796          self.lambda_delta, self.lambda_sat, self._last_logp, self.beta =
796      0.0, 0.0, None, 1.0
797      def set_beta(self, beta: float): self.beta = float(beta)
798      def get_tau(self): return torch.stack([blk.attn.tau_att for blk in
799      self.model.blocks])
800      def step(self, state: GuardianState) -> Dict[str, float]:
801          if not self.enable:
802              return {"lambda_delta": self.lambda_delta, "lambda_sat": self
803      .lambda_sat,
803                      "tau_att": float(self.get_tau().mean().item())}
804          s = state.to_tensor(next(self.policy.parameters()).device).
805      unsqueeze(0)
806          a, logp = self.policy.sample(s); self._last_logp = logp
807          dtau, dl_delta, dl_sat = a[0].tolist(); scale = self.beta
808          with torch.no_grad():
809              for blk in self.model.blocks:
                     tau = blk.attn.tau_att.data + 0.03 * scale * torch.tensor
       (dtau, device=blk.attn.tau_att.device)
```

```
810              overshoot = torch.clamp(tau - blk.attn.tau_max, min=0.0)
811              blk.attn.tau_att.data = (tau - 0.10 * overshoot).clamp
812      (0.3, blk.attn.tau_max)
813          self.lambda_delta = float(np.clip(self.lambda_delta + 0.01 *
814      scale * dl_delta, 0.0, 1.0))
815          self.lambda_sat  = float(np.clip(self.lambda_sat   + 0.01 *
816      scale * dl_sat,  0.0, 0.6))
817          return {"lambda_delta": self.lambda_delta, "lambda_sat": self.
      lambda_sat,
818              "tau_att": float(self.get_tau().mean().item())}
819      def update(self, reward: float):
820          if not self.enable or self._last_logp is None: return
821          loss = -(self._last_logp.mean() * torch.tensor(reward, dtype=
      torch.float32, device=self._last_logp.device))
822          self.opt.zero_grad(); loss.backward(); self.opt.step()
823
824
```

## B.5 CHAOS + TRAINING HEURISTICS (FULL)

Listing 8: Chaos controller and in-loop heuristics.

```
class ChaosController:
    def __init__(self, r: float = 3.9, x0: float = 0.721, amp: float =
    0.25, decay: float = 5e-4):
        self.r, self.x, self.amp0, self.decay, self.t, self._last = r,
    float(x0), float(amp), float(decay), 0, 1.0
    def _amp(self): return self.amp0 * math.exp(-self.decay * self.t)
    def step(self) -> float:
        self.x = self.r * self.x * (1.0 - self.x); self.t += 1
        a = self._amp(); self._last = (1.0 - a) + a * self.x
        return self._last
    def factor(self) -> float: return self._last
    def temp(self, max_extra: float = 0.3) -> float: return 1.0 +
    max_extra * self._amp()

def apply_warm_in(model, step, warm_steps):
    scale = min(1.0, step / max(1, warm_steps))
    with torch.no_grad():
        for blk in model.blocks:
            blk.attn.bias_scale.fill_(scale)

def dropout_glide(model, base_p, phase):
    if base_p >= 0.08:
        tail = max(0.0, 1.0 - (phase / 0.60))
        p_drop = 0.08 + (base_p - 0.08) * tail
        for m in model.modules():
            if isinstance(m, torch.nn.Dropout): m.p = float(p_drop)
```

## B.6 LM MICRO-LOSS (FULL)

Listing 9: Label-smoothed optimization + entropy floor with unsmoothed reporting.

```
def lm_loss_from_xy(cfg, model, x, y):
    logits, stats = model(x)                        # [B,T,V]
    V = logits.size(-1)
    flat_logits, flat_y = logits.view(-1, V), y.view(-1)
    ce_pure_sum = F.cross_entropy(flat_logits, flat_y, reduction="sum")
    ls = getattr(model, "_dyn_label_smooth", cfg.label_smooth)
    ce_sm_sum = F.cross_entropy(flat_logits, flat_y, label_smoothing=ls,
    reduction="sum")
    loss = ce_sm_sum
    if model.training and cfg.R >= 2:
        H_mu = stats.get("mu_entropy", None)
        if isinstance(H_mu, torch.Tensor):
```

```
864          H_max = math.log(max(2, cfg.R))
865          ent_pen = F.relu(cfg.ent_floor_eta * H_max - H_mu)
866          lam_sat = getattr(model, "_lambda_sat", 0.0)
867          loss = loss + (cfg.ent_floor_alpha * 0.5) * (1.0 + lam_sat) *
868       ent_pen
869     return loss, {"ce_pure_sum": ce_pure_sum.detach(),
870               "ce_sm_sum": ce_sm_sum.detach(),
871               "ntok": torch.tensor(y.numel(), device=ce_sm_sum.device
872     )}, stats
```

## B.7 WT2 LOADERS + TS DATASET (FULL)

Listing 10: WT2 loaders (HF/GPT-2 BPE) and sliding-window TS dataset.

```
877   def build_wikitext2_loaders(context_len: int, tokens_per_batch: int,
878                            num_workers: int = DEFAULT_WORKERS,
879       tokenizer_name: str = "gpt2"):
880       datasets = _require("datasets"); transformers = _require("
          transformers")
881       ds = datasets.load_dataset("wikitext", "wikitext-2-raw-v1")
882       tok = transformers.AutoTokenizer.from_pretrained(tokenizer_name,
883       use_fast=True)
884       if tok.eos_token_id is None: tok.add_special_tokens({"eos_token": ""
          })
885       eos_id = tok.eos_token_id
886       def encode_split(split: str):
887           texts = [t for t in ds[split]["text"] if t and not t.isspace()]
888           ids = []
889           for t in texts:
890               ids.extend(tok.encode(t, add_special_tokens=False)); ids.
          append(eos_id)
891           return torch.tensor(ids, dtype=torch.long)
892       train_ids, val_ids, test_ids = map(encode_split, ("train","validation
893       ","test"))
894       batch_size = max(1, tokens_per_batch // context_len)
895       train_ds = RandomChunkDataset(train_ids, context_len)
896       val_ds  = SequentialChunkDataset(val_ids, context_len)
897       test_ds  = SequentialChunkDataset(test_ids, context_len)
898       train_loader = DataLoader(train_ds, batch_size=batch_size, shuffle=
          True,  drop_last=True,
899                            num_workers=num_workers, pin_memory=
900       PIN_MEMORY)
901       val_loader  = DataLoader(val_ds,   batch_size=batch_size, shuffle=
902       False, drop_last=True,
903                            num_workers=num_workers, pin_memory=
904       PIN_MEMORY)
905       test_loader  = DataLoader(test_ds,  batch_size=batch_size, shuffle=
906       False, drop_last=True,
907                            num_workers=num_workers, pin_memory=
908       PIN_MEMORY)
909       return train_loader, val_loader, test_loader, tok.vocab_size, tok

910   class SlidingWindowTS(Dataset):
911       def __init__(self, df: pd.DataFrame, input_cols: List[str],
912       target_cols: Optional[List[str]],
913                   context_len: int, horizon: int, stride: int = 1,
914       normalize: bool = True):
915           self.X = df[input_cols].astype(np.float32).values
916           self.Y = self.X if target_cols is None else df[target_cols].
          astype(np.float32).values
917           self.context_len, self.horizon, self.stride = context_len,
          horizon, stride
          mu = self.X.mean(0, keepdims=True) if normalize else 0.0
          sigma = self.X.std(0, keepdims=True) + 1e-6 if normalize else 1.0
```

```
self.Xn = (self.X - mu) / sigma
if target_cols is None:
    self.Yn = (self.Y - mu) / sigma
else:
    ymu, ysig = self.Y.mean(0, keepdims=True), self.Y.std(0,
keepdims=True) + 1e-6
    self.Yn = (self.Y - ymu) / ysig
self.N = len(self.Xn)
def __len__(self): return max(0, (self.N - (self.context_len + self.
horizon)) // self.stride)
def __getitem__(self, idx: int):
    i = idx * self.stride
    x = self.Xn[i:i + self.context_len]
    y = self.Yn[i + self.context_len:i + self.context_len + self.
horizon]
    return torch.from_numpy(x), torch.from_numpy(y)
```

