# OpenReview forum: "Fuzzy-Gated Training for Transformers: Length-Aware Priors and Gain-Aware Control"
_ICLR.cc/2026/Conference — Submitted to ICLR 2026_

### Official Review · Reviewer_qtQq · 2025-10-25

**Soundness:** 1
**Presentation:** 1
**Contribution:** 1
**Rating:** 0
**Confidence:** 3

**Summary:**

This paper proposes attention modifications to allow for training of seemingly converged small/medium transformers. Specifically, they introduce an attention prior and a RL controller for the softmax temperature. They compare their method to previous work on WikiText-2.

**Strengths:**

* S1: Improving transformer optimization is an important direction.

* S2: The proposed modifications are computationally lightweight and are theoretically motivated.

* S3: Code is provided.

**Weaknesses:**

* W1: The experiments (Table 2) are not sound and don’t allow any inference about whether the proposed method works or not. The reason is that the baselines are not trained under the same experimental conditions. Even if we neglect this, the gains are not convincing; particularly given the very complicated method and could indicate meta-overfitting (the choices are over-optimized).

* W2: The discussion of related work is too small (4 lines only + a couple of lines in the introduction). I encourage the authors to connect and contrast their work to previous work.

* W3: The presentation is very poor: For example, there are paragraphs without any text (e.g., l 238), algorithms are just pasted without any textual descriptions (e.g., l. 74-84), the introduction repeats 3 times the contributions in subsequent paragraphs, parts of the text are hardly understandable (l. 148-150), and it often reads like listing all the things that were done (there’s little to no motivation for the choices). The paper also appears not organized (e.g., the last section is a theoretical section; there’s no conclusion/summary or similar). I’d also like to note that numbers of the proposed method are bolded even though it’s not best (commonly the best method is bolded). There are also several typographic errors.

* W4: Only experiments on Wiki-Text-2 are provided. Further experimentation is needed: ablations of the introduced modifications.

**Questions:**

* Q1: How are the centers found in listing 1?

* Q2: Why is forecasting briefly described and then written that it isn’t included as part of the submission at the end of the paragraph nor run? (e.g., “We include code/templates for ETT and equities to illustrate application, but did not run or report these experiments in this submission.”, l. 323)

* Q3: Why are no test figures provided (empty column in Table 2)?

---

### Official Review · Reviewer_t7TM · 2025-10-30

**Soundness:** 2
**Presentation:** 1
**Contribution:** 1
**Rating:** 2
**Confidence:** 3

**Summary:**

The paper proposes a suite of methods towards improving the training of small transformers by carefully ensuring that in late stages of training, when learning rate is small, inference gains are preserved. This is done by introducing a length-aware positional basis and a "gain-aware" controller, accompanied by a carefully crafted optimization schedule. Experiments on WikiText-2 validate that the method works.

**Strengths:**

1. The paper is practically motivated by an interesting observation: late-stage training gains are often ignored. To ameriolate this issue, the authors propose a series of training-time optimizations to boost the effect of late-stage gains. This is an important problem to solve, especially in small models where it is crucial to extract the maximum amount of inference time quality.
2. The methods employed are practical and interesting:
    * Adding a positional basis to the logits makes sense intuitively as a way to improve sensitivity to spurioius time inference-time improvements.
    * Using Sinkhorn's algorithm to obtain a doubly stochastic matrix is a nice trick that I have not seen in this context before.
3. The experiment is done methodically and the code is given explicitly, aiding in reproducibility of the observations.

**Weaknesses:**

1. I find the paper is not very well written. The writing is done in an extremely concise, report-like fashion, with the main points obscured by a dense compilation of previously undefined, domain-specific technical terms. I would at least appreciate a section in which the main mathematical tools (Sinkhorn's algorithm, entropic transport etc.), assumptions and their properties are clarified. This would help the reader understand the precise algorithm tools being used.
2. As a continuation to the above point, the contribution of the paper appears very tailored to a specific set of constraints, model and setup that the authors were interested in. The fact that only one dataset (WikiText-2) was tested and only specific values of certain hyperparameters were ascertained tells me that the observations, while interesting, mainly concern this specific (or a very similar) setup. Is there a good reason to believe this program could help small transformer training in general? I believe that the paper would be much stronger if it extrapolated its findings into a set of methods that can be used to improve small model training and inference and *showcased* the effectiveness of its methods through rigorous experiments and ablation studies. In its current form, I am not sure I can trust these methods to work beyond the specific setting.
3. The paper also lack rigor. Mathematically the statements appear impressive, but there are no proofs (eg. Prop 1 and the subsequent Lemma), definitions or a rigorous treatment. This makes it hard for the reader to understand the point of having theorems and lemmata, while also causing confusion. Furthermore, the use of various "heavy-machinery" algorithms (Nash mixtures, sinkhorn) is justified only through rough intuition, which makes it seem like these tools are used mainly because they worked. I'm having a bit of a hard time wrapping my head around the design choices made, and would appreciate a clarification in the rebuttal in case I am misunderstanding something.

Overall this paper lists some interesting findings but could benefit from a rewrite to aid in clarity and rigor, as well as a generalization of its argument.

**Questions:**

See above.

---

### Official Review · Reviewer_hPQy · 2025-10-31

**Soundness:** 3
**Presentation:** 3
**Contribution:** 3
**Rating:** 4
**Confidence:** 2

**Summary:**

This paper addresses late-phase optimization stagnation in small/medium Transformers. It proposes Fuzzy-Gated Training, including Regime–Position Alignment (RPA) for a zero-cost, length-aware attention prior, a lightweight gain-aware Guardian controller, and tail-optimizing HPO. Evaluated on WikiText-2, it reduces validation CE/error without extra inference cost. Contributions: (1) Fuzzy inductive bias via RPA; (2) Gain-aware controller sharpening attention when valid; (3) HPO preserving late gains with ablations.

**Strengths:**

1. It innovatively combines fuzzy regime memberships, entropic transport-based Regime–Position Alignment (RPA), and a gain-aware RL controller to address late-phase optimization stagnation in small/medium Transformers, avoiding limitations of rigid positional priors in prior works while ensuring zero inference cost.
2. It maintains high quality through strict experimental controls (matching compute, parameters, and optimizer across WikiText-2 comparisons), ablation studies isolating each component’s effect, and complete code listings in appendices for reproducibility.
3. It clearly articulates the late-phase optimization problem and explains components, and is significant for enabling small/medium Transformers to retain late gains without extra inference overhead, supporting resource-constrained deployment.

**Weaknesses:**

1. Experimental scope is too narrow—only results on WikiText-2 are reported, with no tests on proposed time-series or equities, limiting generalizability. Running small-scale experiments on one non-LM dataset like ETT would help.
2. Failure mode analysis lacks specifics: vague solutions such as monitoring RPA collapse without thresholds or optimal R ranges. Adding concrete triggers like entropy thresholds and R ablation would improve practicality.
3. Baseline comparisons are weak—uses unoptimized baselines like SFT Pythia-70M instead of SOTA small/medium LMs or late-phase optimization methods. Comparing with tuned baselines like TinyLlama variants would highlight advantages.

**Questions:**

1. Provide small-scale results on one time-series dataset (e.g., ETT) to show if the method works for non-language tasks, which clarifies its generalizability.
2. Specify threshold values for membership entropy (to detect RPA collapse) and optimal R ranges, as these details make the method more practical.
3. Explain why SOTA small/medium LMs (e.g., TinyLlama variants) are not used as baselines, and add such comparisons to highlight the method’s advantages.
4. Ablate different R values on WikiText-2 to show RPA’s robustness to hyperparameters.

---

### Official Review · Reviewer_ydZo · 2025-11-01

**Soundness:** 2
**Presentation:** 2
**Contribution:** 2
**Rating:** 4
**Confidence:** 3

**Summary:**

The paper proposes a training-time inductive bias and controller for small/medium Transformers to preserve late-phase gains without changing inference cost. The main component is Regime–Position Alignment (RPA): tokens receive fuzzy regime memberships (Gaussian centers), which are entropically aligned to a length-aware soft-block positional basis via Sinkhorn to produce a normalized, additive attention bias B. The bias is z-scored for stable scaling and added pre-softmax; at test time it purportedly adds no extra parameters/FLOPs. A tiny REINFORCE policy (Guardian) adjusts attention temperature and saturation penalties only when validation gains warrant sharper attention. Additional ingredients include a small context-length “game” (replicator dynamics) and tail-optimizing HPO, plus selective SWA and a nonzero LR floor. The evaluation focuses on WikiText-2 under compute parity, claiming RPA is the primary lever with Guardian/RPA aiding late-phase improvements; inference cost is unchanged.

**Strengths:**

- **Clear intuition**: late-phase progress at small scale is fragile; a learned attention prior and temperature control are plausible ways to preserve it.
- **RPA formulation is interesting**: fuzzy regime memberships aligned to a length-aware basis via Sinkhorn, producing a zero-mean, variance-normalized additive bias that is compatible with softmax invariances.
- **Thoughtful engineering**: z-scoring the bias, warm-in schedules, nonzero LR floor, selective SWA, and load-balancing/entropy regularization for fuzzy routing.

Compute disclosure, training protocol, and code are also specified / provided with good detail.

**Weaknesses:**

- **Empirical support is a bit lacking**: only WT2 is evaluated, and absolute CE/PPL are far above the provided and comparable baselines?.
- **No quantitative ablations**: I would have liked to see experiments isolating RPA vs sinusoidal/relative baselines, Guardian on/off, context game on/off, selective SWA vs standard EMA/SWA, or LR floor choices under strict compute parity with error bars.
- **The zero-cost inference claim is under-specified**: if µ is used to construct B at test time, how is this computed without extra FLOPs? If B is fixed, how is it data-driven at inference?
- **Internal inconsistencies**: Section 10.1 mentions adding log B to logits, elsewhere in Section 2, it seems like only B is added?

**Questions:**

* Can you clarify the zero-cost inference claim? Do you compute µ and B at test time? If yes, quantify FLOPs/latency overhead vs baseline. If no, what is the exact form of the fixed prior applied, and how is it still data-driven?

* Can you clarify some of the details: the main text alternates between adding B vs log B to logits. Which is used in experiments? How sensitive are results to this choice and to z-scoring/clipping?

---

### Meta-Review · Area_Chair_R3PR · 2026-01-06

**Summary:**

The initial reviews are uniformly negative and the authors have not provided a rebuttal.

**Reviewer Concerns:**

No rebuttal provided.

**Reviewer Scores:**

No rebuttal provided.

---

### Decision · Program_Chairs · 2026-01-26

Reject